
# Synthetic Tsunami Waveform Catalogs With Kinematic Constraints

Maria Ana Baptista[1,2], Jorge Miguel Miranda[2,3], Luis Matias[2], Rachid Omira[2,3]

[1]Instituto Superior de Engenharia de Lisboa, Instituto Politécnico de Lisboa, Portugal
[2] Instituto Dom Luiz, Faculdade de Ciências da Universidade de Lisboa, Universidade de Lisboa, Lisboa, Portugal.
5   [3] Instituto português do Mar e da Atmosfera, Lisboa, Portugal

*Correspondence to*: Maria Ana Baptista (mavbaptista@gmail.com)



**Abstract.** In this study we present a comprehensive methodology to produce a synthetic tsunami waveform catalogue in the North East Atlantic, east of the Azores islands. The method uses a synthetic earthquake catalogue compatible with plate kinematic constraints of the area. We use it to assess the tsunami hazard from the transcurrent boundary located between Iberia and the Azores, which western part is known as Gloria Fault. This study focuses only on earthquake-generated tsunamis. Moreover, we assume that the time and space distribution of the seismic events is known. To do this, we compute a synthetic earthquake catalogue including all fault parameters needed to characterise the seafloor deformation covering the time span of 20 kyr, which we consider long enough to ensure the representability of earthquake generation on this segment of the plate boundary. The computed time and space rupture distribution are made compatible with global kinematic plate models. We use tsunami Empirical Green Functions (EGF) to efficiently compute the synthetic tsunami waveforms for the dataset of coastal locations, thus providing the basis for tsunami impact characterization. We present the results in the form of offshore wave heights for all coastal points in the dataset.

Our results focus the North East Atlantic basin showing that earthquake-induced tsunamis in the transcurrent segment of the Azores-Gibraltar plate boundary pose a minor threat to coastal areas north of Portugal and beyond the Strait of Gibraltar. However, in Morocco, Azores, and Madeira islands we can expect wave heights between 0.6m and 0.8 m requiring the evacuation of coastal areas. The advantages of the method are its easy application to other regions and the low computation effort needed.

## 1 Introduction

Deterministic and probabilistic tsunami hazard assessment methods have been developed and applied to many geodynamic contexts and coastal areas. Deterministic assessments are scenario-based approaches, usually considering a maximum credible geological scenario for local and distant sources. They correspond to a "precautionary" approach and are a standard procedure to hazard mapping, warning procedures and evacuation routes for highly sensitive areas, particularly critical infrastructures (González et al, 2009). Probabilistic tsunami hazard assessments (PTHA) providing the evaluation of the annual probability exceedance wave heights and flooding depths (Geist and Lynett 2014), are considered more informative than a single "worst considered case" inundation scenario (LeVeque et al., 2016). In situations where the effects of smaller tsunami events are also important (e.g. interaction with harbours or other coastal structures), probabilistic hazard assessment can give a better insight on expected amplitudes and recurrence times (Geist and Parsons 2006; Power et al. 2007; González et al. 2009; Sørensen et al. 2012, Omira et al., 2015).

Tsunami have several characteristics that differentiate it from other natural hazards. They can inflict significant damage away from the source area, and because of their paucity there is a limited amount of high-quality observations for the same site (Geist and Parsons, 2006). Moreover, it is not possible to describe tsunami propagation using an "attenuation" law-like, and integrate it into a formal probabilistic calculation.

Probabilistic seismic hazard analysis (PSHA) was established by Cornell (1968) to complete the information that can be retrieved from catalogues using geological and geophysical information on the potential earthquake sources. Seismic events are supposed to be described by a Poisson process, and magnitudes are supposed to follow a Gutenberg and Richter (GR) doubly truncated distribution (Kijko and Graham, 1998). This approach is widely used to assess seismic hazard, particularly in areas where historical earthquake catalogues are rather incomplete. However, when the seismogenic area corresponds to a distinct plate boundary, it is possible to use the parameters given by global kinematic plate models to infer the long-term average slip on the fault and constrain the earthquake catalogue. Once the synthetic earthquake catalogue is computed, it is possible to use numerical tsunami modelling to infer the resultant tsunami waveform catalogue in an area of interest.



In this study, we define as seismic significant event, the earthquakes of the catalogue with magnitude greater than 6, being treated as a potential tsunami sources. We consider the simplest case of a rectangular fault with constant slip, and we use scaling laws to infer the fault parameters. These parameters are then used to compute the seafloor deformation using the half space elastic approach (Okada, 1985). Assuming the incompressibility of the seawater we can directly transfer the sea

bottom deformation to the water surface and use it as initial condition for the tsunami propagation and use standard shallow water numerical modelling to provide an accurate assessment of the potential impacts close to the coasts.

The method presented here comprises three steps. The first is the computation of an earthquake synthetic catalogue compatible with long-term plate velocities covering a long-time span – 20kyears - to ensure realistic evaluations of seismic and tsunami hazard. The second is the calculation of tsunami waveforms in selected coastal points for all events in the

catalogue. The third is the post-processing of the use of synthetic tsunami waveforms to infer the impact along the coast. The computation of tsunami waveforms along the coast is a time-consuming process that requires substantial computational resources. To overcome this difficulty, we use the Empirical Green Functions to compute the tsunami waveforms (see Tonini et al., 2016 and references herein) with rectangular prisms as described in Miranda et al. (2014). The use of a representative earthquake catalogue together with the fast computation of tsunami waveforms constitutes an efficient method to evaluate

tsunami impact at discrete locations along the coast.

## 2 Plate kinematic context

The western segment of the Eurasian-Nubian plate boundary extends from the Mid-Atlantic Ridge in the Azores towards the Strait of Gibraltar. This plate boundary is active since ~27Ma, when Iberia became attached to Eurasia, triggering the development of the Azores triple junction. Spreading velocities at the Mid-Atlantic Ridge (see figure 1) are larger for

Eurasia-North America than for Nubia-North America. The plate boundary corresponds broadly to a dextral transcurrent relative motion, where there is an age difference, between a younger "Eurasian" lithosphere, to the north, and an older "Nubian" lithosphere to the south (Luis and Miranda, 2008).

The geometry of this plate boundary was studied since the early years of plate tectonics (e.g. McKenzie, 1972). Between -24ºE and -20ºE, the plate boundary is supposed to follow a prominent morphological feature, the Gloria Fault (see figure 1).

This fault, firstly mapped by Laughton et al. (1972) is a 400-km long fracture, with a vertical offset of several hundreds of meters (this is the Gloria Fault s.s.). Between -20ºE and -18ºE it changes strike, from N84E to N71E (Baptista et al., 2016). East of -18ºE the strike of the main structure changes to N98E, joining the Tore Madeira Rise close to -15.5ºE. Here, there is not a unique fracture zone that could be interpreted as the plate boundary. East of the Tore Madeira Rise the geometry of the Eurasia-Nubia plate boundary is still a matter of debate.

Current plate kinematic models show that both the first and the third segments of the plate boundary follow very closely an inner circle of the Nubia-Eurasia relative plate velocity, with an estimated value of 4 mm/year (e.g. Fernandes et al., 2003 and references herein). Tectonically, both segments can be interpreted as dextral transcurrent faults with minor compressive or normal components. Nevertheless, during the XX century three earthquakes of magnitude greater than 7 occurred in this plate boundary, all of them tsunamigenic: The 8[th] May 1939 $M_S$ 7.1 (Reis et al., 1939), the 25[th] November 1941 M 8.3

(Baptista et al., 2016), and the 26 May 1975 1975 M 7.9, (Buforn et al. 1988; Kaabouben et al., 2008). The 1939 earthquake and tsunami had its source close to the Azores, at -24.34º E, 37.17ºN, on the western tip of the Gloria Fault (Buforn et al., 1988, Reis et al., 2016). The 1941 earthquake and tsunami, one of the largest submarine strike slip events ever recorded, had its source at -19.04º E, 37.41º N, close to the middle of the plate boundary (Udias et al. 1976; Lynnes and Ruff 1985, Baptista et al., 2016). The 1975 earthquake had its epicentre close to -17.50˚ E, 35.90˚ N, on the eastern part of the plate

boundary, 200 km to the south of Gloria Fault. The spatial distribution of these three events shows that there is not a single discrete structure running from the Azores to Gibraltar (see figure 1), which can be considered as the locus of seismic slip.





To compute the synthetic earthquake catalogue, we assume a simplified interpretation model with three homogeneous sections. The first one, between -24ºE and -18.5ºE, to the west of the 1941 earthquake epicentre named segment GF1. To the east of GF1 we consider a split into two subparallel segments GF2 and GF3 (see figure 2). These two sub-segments might correspond to the reactivation of previous transforms faults associated with the Mid-Atlantic Ridge; the northern segment

follows the shape of the Gloria Fault, and the southern segment corresponds to the most probable location of the 26 May 1975 event (Lynnes and Ruff, 1985 and Kaabouben et al., 2008). Figure 2 depicts this model.

### 3. Computation of the synthetic earthquake catalogue

### 3.1 Scaling Laws for Tsunamigenic Sources in the Gloria Fault

In this study, we assume that the co-seismic displacement can be described by the half space elastic approach following

Okada (1985). We use nine independent parameters: fault width $W$, fault length $L$, fault azimuth (strike) $\phi$, dip angle $\delta$, average slip along the fault $u$, slip angle (rake) $\lambda$, and depth below seafloor $h$ to describe each seismic event of the synthetic catalogue. Here, "epicentre" is the geometrical centre of the rectangular fault, which adds two parameters to the list above, $f_{lat}$ and $f_{lon}$. The seismic moment $M_0$ is related with the fault area $A = WxL$, and the average slip $u$, by:

$$M_0 = \mu A u \qquad (1)$$

The relation between the moment magnitude $M_W$ and the seismic Moment $M_0$ is given by equation (2) (Hanks and Kanamori,

15  1979):

$$M_W = \frac{2}{3}(logM_0 - 9.05) \qquad (2)$$

In equations (1) and (2), $\mu$ is the modulus of rigidity. Given the moment release and the rupture area, the stress drop $\Delta\sigma$ can be inferred by:

$$\Delta\sigma = \frac{CM_0}{\Lambda W L} \qquad (3)$$

In equation (3) C and $\Lambda$ are constants that depend on the fault mechanism and geometry. For rectangular strike-slip faults the most appropriate constants for C and $\Lambda$ are given by $C = 2/\pi$; $\Lambda = W$ (Kanamori and Anderson, 1975).

Assuming that all events of the catalogue occur inside the same tectonic area, the dip, the strike and rake angles and the modulus of rigidity can be considered as independent of the magnitude. We describe these parameters by random variables characterized by their mean and standard deviation, with a normal or log-normal distribution: $(\overline{\delta}, \sigma_\delta)$, $(\overline{\phi}, \sigma_\phi)$, $(\overline{\lambda}, \sigma_\lambda)$, $(\overline{\mu}, \sigma_\mu)$. The length, width and slip of the fault constrain the magnitude and can be considered as random variables: $[\overline{W}(M_w), \sigma_w]$, $[\overline{L}(M_w), \sigma_L]$, $[\overline{u}(M_w), \sigma_u]$. The relations between these three parameters are given by equations (1) and (2),

and we assume that slip on the fault is the parameter to be adjusted. Besides, we assume a dominant strike slip regime with a maximum magnitude estimated in 8.5 (the 25 November 1941 event has an estimated magnitude $M_s$=8.2 to 8.4 (Gutenberg and Richter, 1949, Udias et al, 1976). Considering that we have oceanic lithosphere with a typical oceanic crust (7 km thick) and very few or no sediments we will use $5.0x10^{10}Pa$ for the shear modulus ($\mu$). Table 1 presents a summary of the main seismotectonic parameters related to the two largest instrumental events in this plate boundary.

To obtain the scaling relationship between the magnitude, the fault dimensions and the average slip, we analysed the Wells and Coppersmith (1994) and Stirling et al. (2002) compilations. Considering the characteristics of the Azores-Gibraltar plate boundary, we considered only the faults with pure strike-slip mechanisms. As the first compilation includes incomplete information on subsurface rupture length, we computed an average relationship between surface and sub-surface lengths as: $L_{subsurface} = 1.65 L_{superficial}$, to estimate the length of the fault as the subsurface rupture for all the earthquakes where the

width and average displacement on the fault are also known. Table 2 presents a summary of these results.




There are several undesirable features in the case of the use of Stirling et al. (2002) database. These features are the large variations of the stress drop with magnitude, the large changes of the fault aspect ratio with moment magnitude, and the lack of agreement with the largest events recorded in the area (see table 1). The scaling laws derived from the Wells and Coppersmith (1994), also presented in Table 2, are more satisfactory regarding the comparison with those largest events, but

still, they show large changes in the stress drop with magnitude. These effects may be a consequence of the prevalence of continental strike-slip faults, and Ridge Offset Transforms in both compilations, as they are rheological different from intra-oceanic events.

To circumvent these limitations, we applied the approach developed by Matias et al. (2013) for the tsunamigenic sources in the Gulf of Cadiz, based on the works of Scholz (1982). We start from the scaling relationships from Manighetti et al.

10  (2007):

$$D_{max} = \frac{\alpha L}{2} \ if \ L \leq 2W_{sat}; D_{max} = \frac{\alpha}{\frac{1}{L}+\frac{1}{2W_{sat}}} \ if \ L > 2W_{sat} \tag{4}$$

Here, $D_{max}$ is the maximum displacement on the fault, L is the fault length and $W_{Sat}$ is the saturation width, which we take as the maximum rupture thickness. We consider $D_{max} = 2D$ as in Manighetti et al. (2007), $\alpha = 50$ x $10^{-5}$. Since the oceanic crust age at the Gloria Fault changes from 50 Myr to the West to 120 Myr to the East (Luis and Miranda, 2008), we assume that the 600ºC isotherm depth ranges from 35 km to 52 km, with an average value of 40 km for the seismogenic thickness in the

whole Gloria Fault area (e.g. McKenzie et al., 2005, Geli & Sclater, 2008), thus $W_{Sat}=40 \ km$. We consider that the typical aspect ratio $f = L/W$ close to $4.0$ for fault rupture in the Gloria Fault area.

When $L \leq 2W_{Sat}$, W is obtained from a simple linear relationship (equation 4, case 1 in Table 3). When $L > 2W_{Sat}$, the fault width W must be derived by solving a 3rd degree polynomial equation (case 2 in Table 3). In addition, we know that thermo-mechanical properties of the oceanic lithosphere restrict the seismic rupture to the seismogenic layer (or a little more for very

large earthquakes). In our source model, we limit the width of the fault to the seismogenic thickness as discussed above. Therefore, when $L > 2W_{Sat}$ and $W > W_{max}$, the fault width W is obtained by solving a 2nd degree polynomial equation (case 3 in Table 3). Results for the study area are depicted in figure 3, together with the few instrumental large earthquakes, and the Robinson (2011) solution for the 2004 December 23 Mw=8.1 Tasman Sea oceanic intraplate strike-slip event, which can be considered a comparable event.

The magenta curve in figure 3 depicts the final scaling law between fault displacement and fault length. Scaling laws for width, length and average slip are depicted in table 3. By using this scaling law there is a high but nearly constant stress drop and a constant aspect ratio until the rupture exceeds the seismogenic thickness.

### 3.2 Earthquake Recurrence

Earthquake recurrence for the Azores Gibraltar transcurrent plate boundary cannot be assessed from historical information or

instrumental catalogues, due to the location of plate boundary with respect to the neighbouring land areas, and the resulting incompleteness of catalogues. Nevertheless, one may consider that the time distribution of earthquakes can be described by a truncated GR law, with cumulative distribution function given by (Kijko and Graham, 1998):

$$F_M(m|m_{min}, m_{max}) = 0 \ if \ m < m_{min}; \frac{1-e^{-\beta(m-m_{min})}}{1-e^{-\beta(m_{max}-m_{min})}} \ if \ m_{min} \leq m \leq$$
$$m_{max}; 1 \ if \ m > m_{max} \tag{5}$$

Here $m_{min}$ is the minimum magnitude of interest, $m_{max}$ is the maximum earthquake magnitude for the Gloria Fault, and $\beta = b\ln(10)$, being b the usual GR parameter. Assuming a Poisson distribution, and taking $\lambda$ as the yearly mean

occurrence of earthquakes with a magnitude greater than the threshold value $m_{min}$, the mean rate of occurrence of earthquakes with magnitude equal or greater than m ( $m_{min} \leq m \leq m_{max}$ ) is given by:


$$\dot{N}(m) = \lambda[1 - F_M(m|m_{min}, m_{max})] \qquad (6)$$

Considering the segmentation of Gloria Fault as described in section 1, the lengths of GF1, GF2 and GF3 are 455 km, 271 km and 278 km respectively (c.f. figure 2). We consider that plate kinematic strain is fully coupled to GF1 section but split in two between sections GF2 and GF3. We deduce the maximum magnitude $m_{max}$ of each section from the previous scaling laws. The maximum magnitudes are 8.5, 8.1 and 8.1 for segments GF1, GF2 and GF3 respectively, corresponding to the

rupture of the entire segment. We assume that the faults are vertical with a maximum width of 40 km. We consider a modulus of rigidity of $\mu=5.0 \times 10^{10}$ Pa, a seismic coupling coefficient between plate kinematics and earthquake generation $\chi=1.0$ for GF1 and $\chi=0.5$ for GF2 and GF3, and $b(\beta) = 0.98$, as estimated by Bird & Kagan (2004) for slow oceanic transform faults. Considering $m_{min}=6.0$ we can estimate earthquake activity rate $\lambda$.

We generated the seismic synthetic catalogue for a total time span of 20 kyr, using the MATLAB code provided by Andrzej

Kijko (Kijko & Graham, 1998). We compared the catalogue with the Gutenberg-Richter model that generated it, and the summing up of the total moment generated was compared with the nominal slip rate as given by plate kinematic models (~4 mm/year, e.g. Fernandes et al., 2003). Different random generations of the synthetic catalogue showed significant fluctuations in the number of earthquakes generated for large magnitudes and, consequently, in the total slip rate. We compensated these differences by extending or reducing the time span of the catalogue. We made several runs to obtain

coherence with the GR model and with the nominal slip rate. Results are shown in Figure 4.

Given the time sequence of earthquakes in the synthetic catalogue for the whole area, we need to attribute each event to the best location, ensuring compliance with the kinematic constraints. To do this, we use a rectangular model of the fault plane divided into small elements $1 \times 1$ km$^2$ (inset of Figure 5). The slip on the fault is initialized with a smoothed random function, with zero average (Figure 5, above). Next, for each event in the catalogue, we search all possible locations for the fault

rupture and select the one with the smallest total slip. Once the location is selected, the corresponding slip on the rectangular rupture area is applied and we proceed until the whole catalogue is exhausted. The success of the procedure can be evaluated by the final cumulated slip on the fault segments shown in Figure 5. The colour scheme used emphasizes the irregular slip distribution that results from the methodology. However, when compared to the nominal slip, we see that extreme values are limited in space and not systematically distributed, as would be expected from a uniform random distribution of earthquakes.

To complete the parameters needed for the computation of the tsunami source we use the mean geometrical parameters of Gloria Fault. For each event we compute the epicentre coordinates, strike, dip and rake, using a Gaussian random generator. The standard deviations are the following: epicentre coordinates: ±0.1°, fault strike ±5°, fault dip = 85±5°, slip rake = 170±5°. Figure 2 shows the final epicentre distribution.

**4. Tsunami waveform simulation**

The use of empirical Green functions to compute the tsunami waveforms associated with a given earthquake source in a collection of points of interest (POIs) presented by Miranda et al (2014) includes the following four steps. The definition of an area where the initial sea surface displacement is significantly different from zero for all possible sources under study; the partition of the source area into unit prisms of water 1 m high, where the length and width of those prisms are as small as possible to ensure a good description of the initial displaced area; the definition of a set of points of interest (POIs) along the

coast, which correspond to the location where hazard is to be computed, and the computation of the tsunami waveforms at all POIs due to unit prisms, using a linear shallow water model. These four components form the empirical Green Functions Database (GFD) for the study area.

We define as relevant source area the region where the initial sea surface displacement caused by the co-seismic deformation is significantly different from zero (in this case > 10 cm). Considering the close clustering of seismic activity around Gloria

Fault, we defined as relevant source area the rectangular domain $25^0$W - $14^0$W, $35^0$N - $39^0$N. We divided this area into 110


(longitude) times 40 (latitude) unit sources, with 0.1º x 0.1º. The empirical Green functions for each unit source was done using the code NSWING (Miranda et al., 2014). NSWING uses the discretization and explicit leap-frog finite difference scheme to solve the shallow water equations in spherical and Cartesian coordinates developed by Liu et al. (1998). The calculations presented here use spherical coordinates and a time step of 2.5s. The model ran 7200 time steps, corresponding to a Courant Number ~ 0.45 for 18 Points of Interest located along the coasts of Iberia, Africa and the Atlantic islands (see Figure 6 and table 5 for locations).

The synthesis of the tsunami waveform $\eta_m(t_t)$ at the $m$ point of interest for the time span $t_t$ is given by a superposition principle:

$$\eta_m(t_t) = \sum_{i=1}^{nl} \sum_{j=1}^{nc} G_{ij}^m(t_t) h_{ij} \tag{7}$$

where $G_{ij}^m(t_t)$ are the empirical Green's functions computed by the shallow water model for $t$ time steps, (*nl x nc*) are the set of unit sources and $h_{ij}$ is the amplitude of the initial water displacement attributed to the $ij$ unit source. The computation of the synthetic tsunami waveforms at each PoI comprises the selection of the source parameters from the seismic catalogue, the computation of the corresponding elastic deformation of the seafloor $h_{ij}$, and the use of equation (7) to synthetize the tsunami waveforms $\eta_m(t_t)$.

## 5. Tsunami Hazard Assessment

Tsunami hazard is computed in terms of wave height at each POI. Because we ran a linear approach and the depths of the different points of interest are variable, we use Green's Law (Green, 1837) that expresses the conservation of wave energy flux, to reduce all amplitudes to comparable depths.

$$\frac{\eta_s}{\eta_d} = \left(\frac{H_d}{H_s}\right)^{1/4} \tag{8}$$

where $\eta_s$ and $\eta_p$ represent the wave amplitudes in shallow and deep water and $H_s$ and $H_d$ are the corresponding to 30 m water depth.

In Figure 6, the maximum amplitude at each POI is plotted against the event number in the catalogue. Results show that the largest amplitudes occur in the Azores (Ponta Delgada, see figure 1) and Madeira Island. These results are compatible with the maximum observed amplitudes of the 25th November 1941 tsunami (Baptista et al., 2016). Along the Western Iberian coast, the wave heights decrease fast north of Douro (Portugal) with values that do not exceed 20 cm in Vigo (Spain). However, South of Douro and along the southern Portuguese coast (Vila-do-Bispo to Vila Real de Santo António) the wave heights can reach up to 60 cm. In the Atlantic coast of Morocco wave heights between Essaouira and Casablanca can reach 0.5 m decreasing to less than 0.2 m towards Safi (see figure 1 for locations).

In figure 7 we present the estimated relative frequency of tsunami wave heights for four Points of Interest used in this study. These points are Ponta Delgada in the Azores Islands, the western PoI of the grid, Porto Santo in Madeira archipelago, Casablanca in Morocco and Leixões in northern Portugal (these PoIs are named PDE, PRS, CSB and LXS respectively in figure 1). These four points are located west, south, southeast and northeast of the source thus giving an azimuthal overview of the study domain. The results presented in figure 7 show that the highest amplitudes of 0.8 m occur at Azores and Madeira Islands. However, these values correspond to low frequency events (~$10^3$ yrs). Towards east the amplitudes are smaller not exceeding 0.5 m.





**6. Discussion and Conclusions**

The results presented in figure 6 show that the tsunamis from Gloria fault pose minor threat to the northern part of the North East Atlantic basin (north of Douro) and towards the Strait of Gibraltar. Along the South Portuguese coast, from Vila do Bispo to Vila Real de Santo António. Also, the amplitudes in southern Spain (Huelva and Cadiz) do not exceed 40 cm. The

potential impacts are limited to the interaction with harbours. In the Azores (Ponta Delgada) and Madeira (Funchal and Porto Santo) wave heights can reach 80 cm, leading to the need of evacuation of low coastal areas in case of an event. Along the Moroccan coast the wave amplitudes are maximum in Casablanca decreasing towards south. Sea level rising in the next decades, because of global warming, will add to these amplitudes thus corresponding to a larger hazard, particularly in shallow areas. These results agree with what is known from the historical record. Strike slip tsunamis generated at the

Azores Gibraltar plate boundary were observed during the XX Century but with limited impact on the neighbouring coastal areas (Baptista and Miranda, 2009, Baptista et al., 2016). The largest impact was observed during the 1941 event, associated with the harbour of Leixões in the north of Portugal.

The approach described here is easy to implement in all geological environments where there is enough information to compute a synthetic earthquake catalog. The case study presented here corresponds to a favourable situation were the fault

geometry is supposed to be simple, there is a well constrained long term tectonic slip, and there are several large instrumental tsunami events, which can be used to challenge the synthetic results obtained by numerical modelling. When dealing with geological situation of increased complexity there will be the need for additional *a priori* constraints for the establishment of the catalog of events.

In this study, we assume perfect seismic coupling ($\chi$=1.0). Other choices could be made. Bird and Kagan (2004) proposed

for slow oceanic transform faults (u<39 mm/y) that $\chi$~0.93 for a seismogenic thickness of 14 km. Since the synthesis of the tsunami waveforms equation (7) is linear, a different seismic coupling parameter can be easily accommodated in the hazard curves by considering an appropriate time span for the synthetic catalogue. The size of the initial deformation of the seafloor is dependent on the value of the rigidity. For the case of the Azores Gibraltar plate boundary rigidity has been differently evaluated in a significant way: Johnston, (19969 uses $\mu=6.5x10^{10}\ Pa$ (e.g.). Since $\mu = \rho\,V_S^2$ Stich et al. (2007) proposes a

slightly higher value $\mu=7.0x10^{10}\ Pa$. Heinrich et al. (1994) assumes a lower value, $\mu=4x10^{10}$ Pa. The choice of this parameter will influence the result in terms of tsunami hazard.

The numerical modelling of tsunami waveforms based on the use of empirical Green's function is a simple and efficient approach. The direct determination of a catalog of tsunami waveforms provides a direct description of the hazard, which is described as an empirical histogram of the frequency of occurrence of a given tsunami water height. Information concerning

expected frequencies or the incorporation of the effect of tides can easily be added to this methodology, providing planners and emergency managers with the information they need to develop tsunami resilience of the coastal populations.

Finally, it is worth pointing out that the instrumental observations in the XX century correspond to the realization of low frequency seismic and tsunami events.

**Acknowledgements**

The authors wish to thank Dr. Kijko for providing the MATLAB code and the reviewers for their suggestions to improve this manuscript. This work was funded by projects ASTARTE – Assessment Strategy and Risk Reduction for Tsunamis in Europe Grant 603839 – FP7.



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

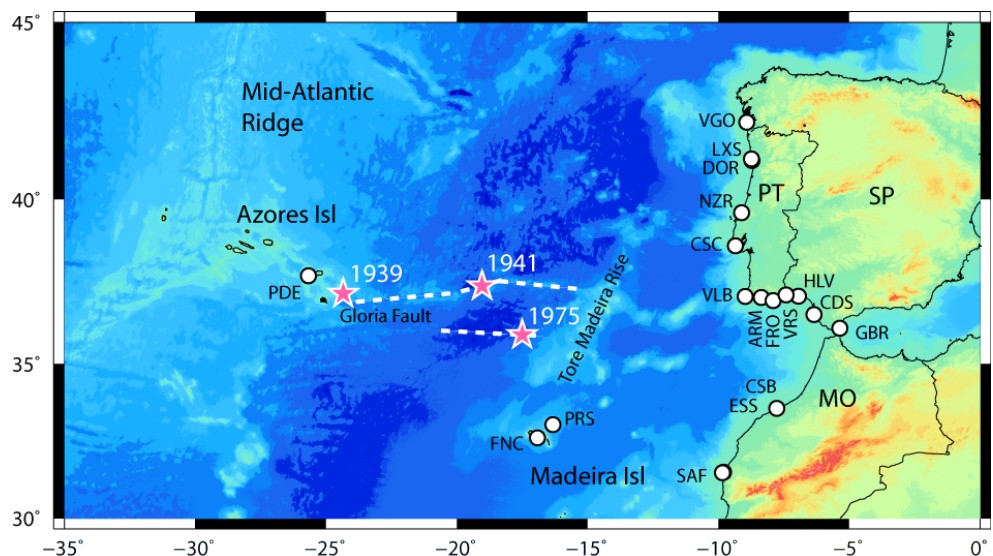

**Figure 1 Shaded bathymetry map of Iberia, northwest Africa and Central Atlantic. The three red stars correspond to the epicentres of the large strike slip events in the study area (see text for details). The Points of Interest used in this study are: VGO**
20   **(Vigo), LXS (Leixões), DOR (Douro), NZR (Nazaré), CSC (Cascais), VLB (Vila do Bispo), ARM (Armação), FRO (Faro), VRS (Vila Real de Santo António), HLV (Huelva), CDS (Cádis), GBR (Gibraltar), CSB (Casablanca), ESS (Essaouira), SAF (Safi), PRS (Porto Santo), FNC (Funchal), and PDE (Ponta Delgada). PT (Portugal); SP (Spain); MO (Morocco).**

25





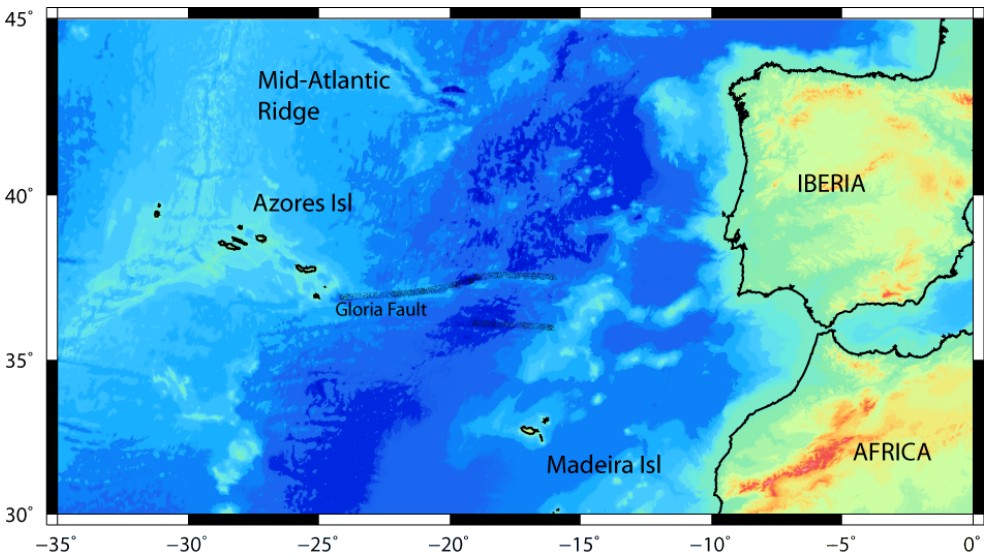

**Figure 2: Decomposition of Gloria Fault into three main segments: GF1, GF2 and GF3. The black dots correspond to the epicentre distribution as given by the seismic catalogue.**





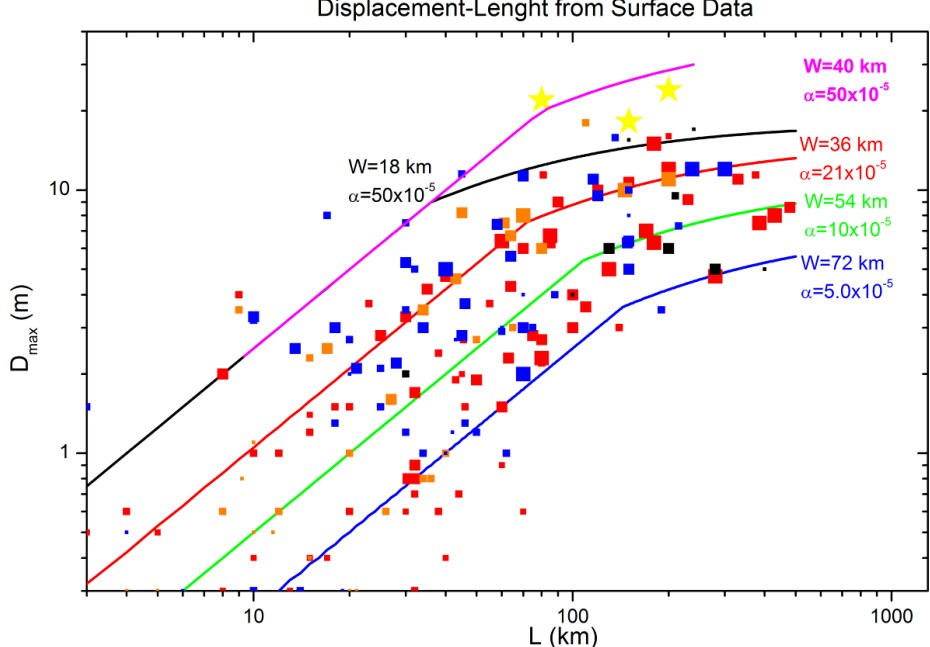

**Figure 3: Maximum displacement versus fault length derived from surface measurements (adapted from Manighetti *et al*. 2007): red squares, strike-slip events; orange squares, dip-slip events; blue squares, composite faults. The black squares were not used in the regression analysis. The size of the symbols is proportional to the quality of the data. Also represented are the 4 typical laws selected by the automatic regression of Manighetti *et al*. (2007; coloured lines). The data points for the Gloria Fault and Tasman Sea earthquakes are plotted as yellow stars, and the magenta line depicts the adopted scaling law (see text for details).**





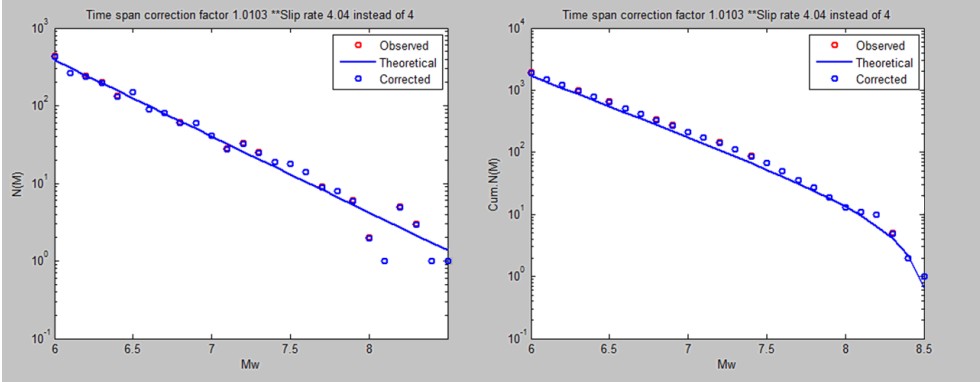

**Figure 4: Comparison of the adopted synthetic catalogue generated for GF1 section and the GR model (blue line). The catalogue values are in red. The blue circles show the corrected number of earthquakes to attain the nominal slip rate.**





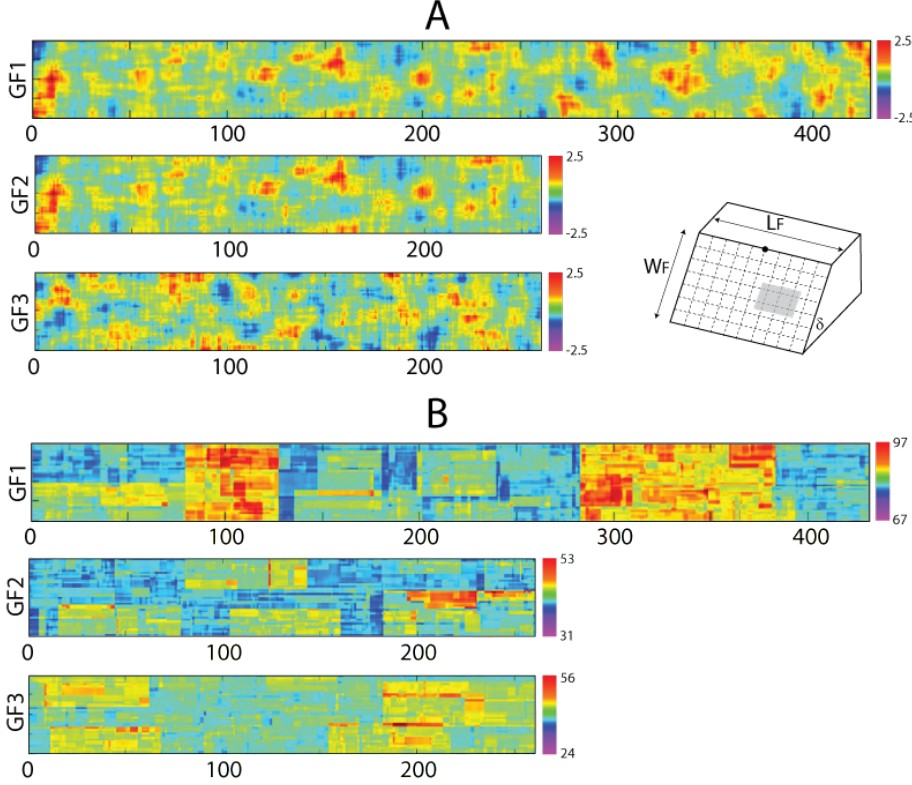

**Figure 5: A: Initial slip on the Gloria fault active sections (GF1, GF2 & GF3) after a smoothed random slip generation, with zero mean. B: Final slip on the Gloria fault sections after 20000 years of s synthetic catalogue. Minimum, maximum and nominal cumulated slips are (in m): GF1 (67 to 97, 80); GF2 (30 to 53, 40); Gf3 (24 to 55, 40).**





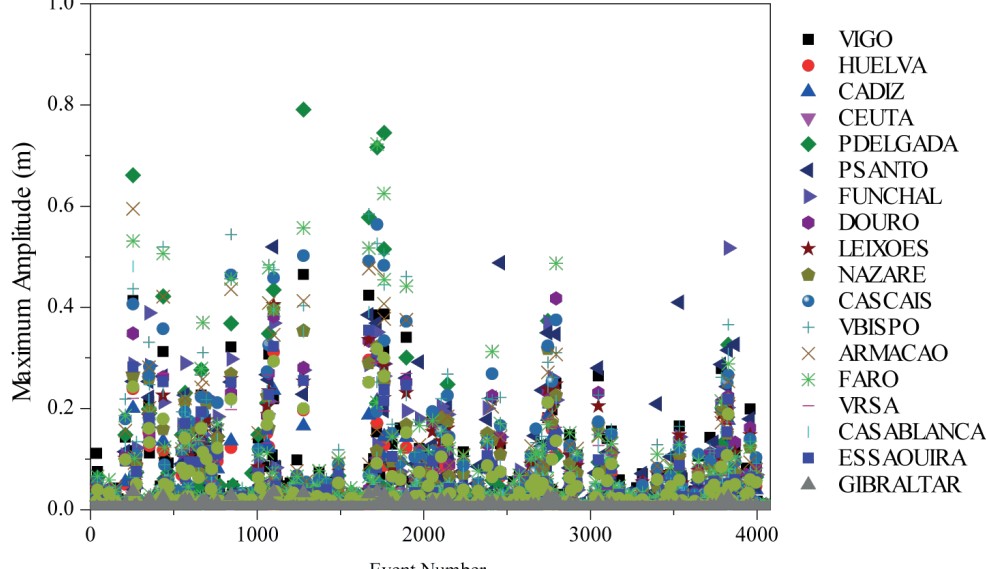

**Figure 6: Time distribution of tsunami wave heights at the Points of Interest, for the entire catalog. The maximum amplitude computed by the shallow water model reaches 80 cm in Ponta Delgada (Azores). Significant amplitudes are also found at Madeira (Funchal and Porto Santo).**





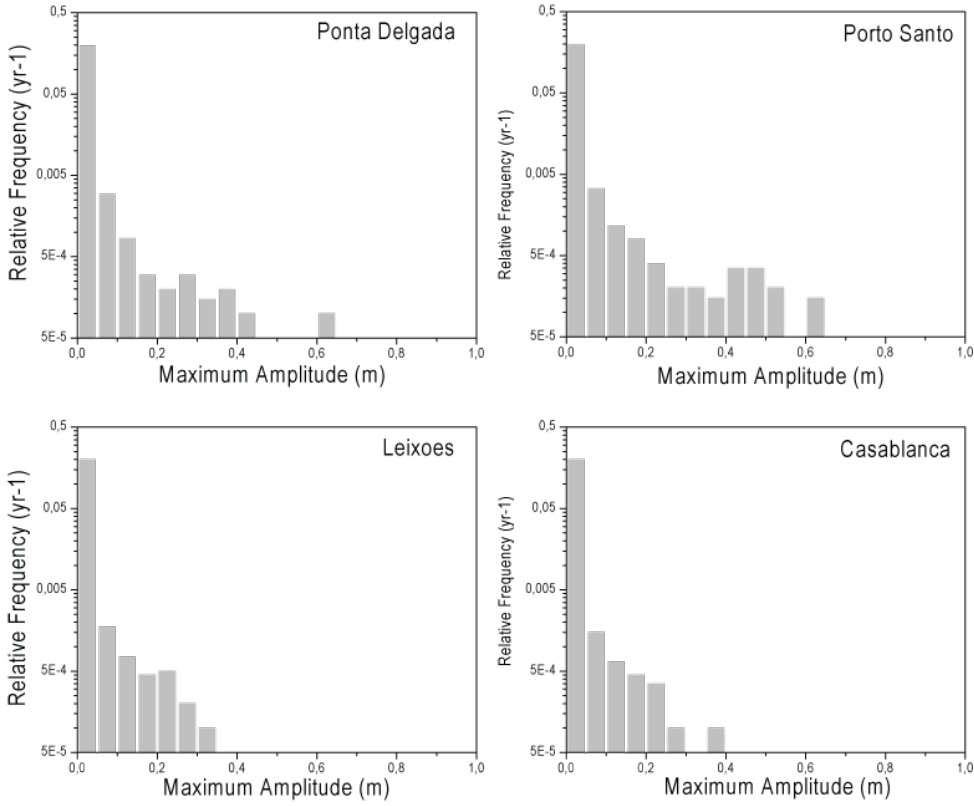

**Figure 7: Estimated relative frequency of tsunami wave heights for four Points of Interest used in this study where the impact from Gloria sources is meaningful (see figure 1 for locations).**





| Event | $\mu$ $10^{10}$ Pa | $M_w$ | L km | W km | u m | $\Delta\sigma$ MPa | Strike (°) | Dip (°) | Rake (°) | Reference |
|---|---|---|---|---|---|---|---|---|---|---|
| 1941 | 3.0 | 8.3 | 200 | 45 | 12 | 5.6 | 76 | 88 | -161 | Udias et al., 1976; Lynnes and Ruff, 1985, Baptista et al., 2011 |
| 1941 | 5.0 | 8.4 | 200 | 45 | 12 | 9.6 | 76 | 88 | -161 | Modified from above according to the proposed higher $\mu$ |
| 1975 | 4.0 | 7.9 | 80 | 20 | 11 | 14 | 288 | 72 | -176 | Buforn et al., 1988; Argus et al., 1989. |

Table 1 - **Summary of the main seismotectonic parameters related to the two largest events occurred during instrumental times in the Gloria Fault area.**





| $M_w$ | $M_0$ | Stirling et al. (2002) | | | | | | Wells& Coppersmith (1994) | | | | | |
|---|---|---|---|---|---|---|---|---|---|---|---|---|---|
| | | L | W | u | A | L/W | Δσ | L | W | u | A | L/W | Δσ |
| N/A | Nm | Km | km | m | Km2 | N/A | MPa | km | km | m | Km2 | N/A | MPa |
| 6.50 | 6.31X10$^{18}$ | 32.7 | 10.4 | 0.37 | 339 | 3.14 | 1.1 | 32.8 | 11.9 | 0.29 | 391 | 2.76 | 0.98 |
| 6.75 | 1.50X10$^{19}$ | 48.8 | 11.6 | 0.53 | 567 | 4.21 | 1.5 | 45.7 | 12.8 | 0.51 | 583 | 3.59 | 1.45 |
| 7.00 | 3.55X10$^{19}$ | 72.8 | 13.0 | 0.75 | 947 | 5.60 | 1.8 | 63.7 | 13.7 | 0.88 | 869 | 4.66 | 2.16 |
| 7.25 | 8.41X10$^{19}$ | 108.5 | 14.6 | 1.06 | 1582 | 7.43 | 2.3 | 88.6 | 14.6 | 1.53 | 1296 | 6.06 | 3.21 |
| 7.50 | 2.00X10$^{20}$ | 161.7 | 16.3 | 1.51 | 2644 | 9.92 | 3.0 | 123.4 | 15.7 | 2.66 | 1933 | 7.88 | 4.76 |
| 7.75 | 4.73X10$^{20}$ | 241.1 | 18.3 | 2.14 | 4417 | 13.17 | 3.7 | 171.8 | 16.8 | 4.63 | 2883 | 10.24 | 7.07 |
| 8.00 | 1.12X10$^{21}$ | 359.4 | 20.5 | 3.05 | 7381 | 17.53 | 4.7 | 239.2 | 18.0 | 8.03 | 4298 | 13.31 | 10.50 |
| 8.25 | 2.66X10$^{21}$ | 535.8 | 23.0 | 4.32 | 12332 | 23.30 | 6.0 | 333.0 | 19.2 | 13.96 | 6410 | 17.30 | 15.58 |
| 8.50 | 6.31X10$^{21}$ | 798.8 | 25.8 | 6.12 | 20606 | 30.96 | 7.6 | 463.7 | 20.6 | 24.24 | 9558 | 22.49 | 23.14 |

5   **Table 2: Scaling laws for the width, length and average slip derived from the compilations by Stirling et al. (2002) and Wells & Coppersmith (1994).**





| Mw | $M_0$ (Nm) | L (km) | W (km) | u (m) | A (km2) | L/W | $\Delta\sigma$ (MPa) | case |
|---|---|---|---|---|---|---|---|---|
| 6.50 | $6.31 \times 10^{18}$ | 16.6 | 4.15 | 2.08 | 69.0 | 4.0 | 15.9 | 1 |
| 6.75 | $1.50 \times 10^{19}$ | 22.1 | 5.54 | 2.77 | 123 | 4.0 | 15.9 | 1 |
| 7.00 | $3.55 \times 10^{19}$ | 29.5 | 7.38 | 3.69 | 218 | 4.0 | 15.9 | 1 |
| 7.25 | $8.41 \times 10^{19}$ | 39.4 | 9.8 | 4.92 | 388 | 4.0 | 15.9 | 1 |
| 7.50 | $2.00 \times 10^{20}$ | 52.5 | 13.1 | 6.57 | 690 | 4.0 | 15.9 | 1 |
| 7.75 | $4.73 \times 10^{20}$ | 70.0 | 17.5 | 8.75 | 1226 | 4.0 | 15.9 | 1 |
| 8.00 | $1.12 \times 10^{21}$ | 96.5 | 24.1 | 10.9 | 2329 | 4.0 | 14.4 | 2 |
| 8.25 | $2.66 \times 10^{21}$ | 138 | 34.5 | 12.7 | 4769 | 4.0 | 11.7 | 2 |
| 8.50 | $6.31 \times 10^{21}$ | 239 | 40.0 | 15.0 | 9559 | 6.0 | 11.9 | 3 |

**Table 3 - Scaling laws for the width, length and average slip derived by a semi-empirical approach.**





|  | $M_{min}$ | $M_{max}$ | L (km) | W (km) | b | $\lambda$ (y$^{-1}$) | $\dot{u}$ (mm y$^{-1}$) |
|---|---|---|---|---|---|---|---|
| GF1 | 6.0 | 8.5 | 455 | 40 | 0.98 | 0.095 | 4.0 |
| GF2, GF3 | 6.0 | 8.1 | 275 | 40 | 0.98 | 0.054 | 2.0 |

**Table 4 - Summary of truncated Gutenberg Richter laws for the active sources in the Gloria Fault area**





| POI | Longitude(º) | Latitude(º) | Depth (m) |
|---|---|---|---|
| Vigo | -08.92 | 42.23 | 009.3 |
| Huelva | -06.95 | 37.09 | 017.5 |
| Cadiz | -06.35 | 36.53 | 019.0 |
| Ceuta | -05.30 | 35.88 | 018.7 |
| Ponta Delgada | -25.66 | 37.71 | 283.8 |
| Porto Santo | -16.32 | 33.05 | 342.5 |
| Funchal | -16.92 | 32.63 | 083.0 |
| Douro | -08.72 | 41.14 | 021.8 |
| Leixões | -08.75 | 41.18 | 028.9 |
| Nazaré | -09.11 | 39.60 | 057.2 |
| Cascais | -09.34 | 38.62 | 012.5 |
| Vila Bispo | -08.97 | 37.09 | 023.2 |
| Armação Pera | -08.37 | 37.05 | 017.3 |
| Faro | -07.91 | 36.96 | 005.4 |
| VilaReal | -07.41 | 37.13 | 014.4 |
| Casablanca | -07.77 | 33.58 | 021.7 |
| Essaouira | -09.81 | 31.51 | 025.0 |
| Gibraltar | -05.36 | 36.10 | 261.9 |

**Table 5 – Points of Interest used in this study**

