# Peer review of "Synthetic Tsunami Waveform Catalogs With Kinematic Constraints"

_Natural Hazards and Earth System Sciences, 2017_

## Referee Comment (RC1) · Anonymous Referee #1 · 27 Mar 2017

Overall

The authors studies the faultiness close to the Iberia peninsula, in order to estimate the tsunami waveforms that could attack the western coasts of Portugal, Spain and Morocco. Generally, their approach appears interesting, yet there are a number of issues that need to be clarified before the paper can be accepted for publication.

Major comments

-It is surprising that no mention is made at all in this paper to the historic Lisbon earthquake and tsunami. Such a major event, and its relation to plate tectonics, should at least be mentioned somewhere in the introduction.

-In section 2 the authors describe three tsunamigenic earthquakes. However, they pro-

vide no indication concerning the records of the inundation heights, etc. This would be of major interest to any readers. Later, the authors state "These results are compatible with the maximum observed amplitudes of the 25th November 1941 tsunami (Baptista et al., 2016)." What exactly are these amplitudes, and exactly how well do they compare?

-Figure 2 is of very poor quality. It is almost impossible to understand what the authors are referring to regarding the dots (this reviewer has very good eyesight, yet had to put his eyes almost an inch away from the screen to notice anything). The authors could use a different colour (maybe red?) for the dots, and clearly label the segments on the figure?

-P7L4 "The model ran 7200 time steps," How long was this in hours? It would be interesting for the authors to show at least one example of the time history of the wave profile at one point along the coastline (say Lisbon or Cadiz, for the worst inundation height) Otherwise, it is difficult to verify how well the author's model is working.

-Why did the authors choose to show the tsunami amplitude at the 30m line? Somehow, this reviewer would have expected the authors to verify that the model they are using can provide realistic inundation heights, at least at one location. Otherwise, how can they really claim that their model reproduces well historical events (essentially, this is an issue of calibration and verification of their model)

-Discussion. It is difficult to accept the hazard analysis presented by the authors, as they only estimated the tsunami wave heights at a depth of 30m. Also, it is strange to talk about sea level rise for the case of tsunamis, and difficult to sea how this would have a great influence, unless the authors are talking about the end of the 21st century (even then...)

Minor comments

-L15 "we can expect wave heights between 0.6m and 0.8 m requiring the evacuation of

coastal areas." , Normally such small heights would produce little to no damage (based on other events). Obviously the coastline should be evacuated, just in case...

- "The computation of tsunami waveforms along the coast is a time-consuming process that requires substantial computational resources." Is this really the case nowadays? Somehow it does not feel that the resources needed are that substantial any more, but it could be that this reviewer is used to how things were over a decade ago...

-In the present work the authors assume a uniform slip. This is a common assumption, though at present some other authors have started to use non-uniform slips. Maybe this can be briefly discussed? Could non-uniform slips result in higher tsunami amplitudes?

-P7L16 This phrase is unclear "Because we ran a linear approach and the depths of the different points of interest are variable," What do the authors mean by "we ran a linear approach"? P8L32 "Finally, it is worth pointing out that the instrumental observations in the XX century correspond to the realization of low frequency seismic and tsunami events." It is not clear what the authors mean. Also, note that mentioning records of only one century with regards to tsunami events is potentially dangerous.

Grammar, typos

Generally speaking the language is ok, though at times there are problems with some sentences. Below in an non-exhaustive list of problems found by this reviewer while going through the document.

L5 "known as the Gloria fault"

L8 "a time span of 20,000 years" (is this what the authors are trying to say?)

L29 "Tsunami have several characteristics that differentiate it from other natural hazards." Maybe "Tsunami waves have several characteristics that differentiate them from other types of natural hazards"

L31 "Moreover, it is not possible to describe tsunami propagation using an "attenuation" law-like," this is confusing, please improve/rephrase this sentence.

P3L1, delete the comma after "event"

P3L8 please use 20,000 years (this reviewer prefers this notation, and in any case there should be a space between k and years)

P3L21 "age difference, between a younger" delete comma after difference

-P5L12 and following. All parameters in the paper must be in italics. Please check these and the entire paper again

-P7L19 change to "the corresponding wave heights at 30 m water depth.

-P8 L4 "Along the South Portuguese coast, from Vila do Bispo to Vila Real de Santo António." Incomplete sentence

---

## Referee Comment (RC2) · J. A. Alvarez-Gómez (Referee) · 11 Apr 2017

Review:

The authors present a probabilistic tsunami hazard estimation methodology based on a synthetic earthquake catalog. They apply the methodology to the North East Atlantic, and specifically to the Gloria Transform Fault generated earthquakes. The tsunami propagation model used is based on Empirical Green Functions (EGF). The authors use EGF to save computational time, using a database of precomputed propagations (110 x 40 – 4400 – unit cell propagations), and adding linearly the needed EGF for each seismic event.

Below I expose the main concerns with the work in more detail, but based on the lack of detailed explanations and discussion on the methodology proposed, the misuse

of some references and concepts, and the total absence of uncertainties estimations (epistemological and aleatory) I cannot recommend the acceptance of the work on its present form, and major revisions are needed.

In brief the expected major revisions are:

The structure of the text must be improved. The work presents a new methodological approach but seems a case study for the Gloria Transform Zone. I suggest to present first the methodological approach from a theoretical point of view, and then an example of application to the Gloria Transform and its results.

The authors should revise the use of scaling relations and bibliographical data to obtain the fault parameters. They should incorporate the associated aleatory uncertainties on their calculations, and a logic-tree approach if needed for some parameters. The influence of the selections made in the results should also be discussed.

The tectonic setting and assumptions done is key for the synthetic earthquake catalog. The authors should show in a figure the kinematic constrains and the modeled faults.

The authors should incorporate an estimation on the accuracy of the tsunami simulation results comparing their propagation results to some historical event. As there are three tsunamigenic historical events (as described in the text) the authors could choose one of them to estimate the accuracy of the model. This is specially relevant as the model of Miranda et al. 2014 has not been validated.

Finally, the references should be revised as they are misused, omitted and/or forced to support affirmations not always present in the original work.

Main concerns:

In Table 1 the 1975 earthquake data is referred to Buforn et al., 1988 and Argus et al., 1989. The latter does not describe the 1975 event, and does not provide any seismo-tectonic data, and the parameters shown for this event in Table 1 does not coincide with the parameters of Buforn et al. (1988). The parameters for the 1941 event does

not coincide with the data provided by Udias et al. (1976) or Lynes and Ruff (1985). The Baptista et al. (2011) is a reference to a conference communication abstract so I suppose that they simply use the original Udias et al. (1976) focal mechanism parameters.

The scaling relations used in the work are the Wells & Coppersmith (1994) and Stirling et al. (2002). Both works are based on surface ruptures in continental lithosphere. I recommend the use of other scaling relations that already incorporates oceanic lithosphere events like Blaser et al. (2010) or Allen et al. (2017) relations. These relations use the subsurface rupture length, so there is no need to use the ad-hoc relation of Length – Subsurface Length used by the authors. The relation of Manighetti et al. (2007) is used but forced to fit the dimensions assumed for the earthquakes in the zone; if different dimensions are used, as for example the 200 km proposed by Buforn et al. (1988) for the 1975 earthquake, then the Dmax diminishes and then would fit without forcing the Manighetti et al. (2007) relation. There are inconsistencies in the use of relations and dimensions, and being this aspect crucial to the tsunami generation it should be adequately justified and discussed and an error estimation on the results should be presented.

The concept of seismic coupling is misused in section 3.2; the seismic coupling is the proportion of tectonic slip produced seismically vs aseismically. If GF1 is fully coupled it is reasonable to assume that GF2 and GF3 are also fully coupled, then the seismic coupling should be also 1. The authors in fact are talking about a slip distribution between two parallel faults, absorbing each the 50% of the total tectonic slip. This should be clearly explained and avoid the confusion with the concept "seismic coupling".

The authors use a generic b-value of 0.98 from a global analysis (Bird and Kagan, 2004) and I wonder why they do not simply calculate the b-value from the seismic catalog of the area, which is the appropriate approximation. Moreover, the authors do not explain how they obtain the earthquake activity rate in addition to the b-value. In

general the method to generate the synthetic earthquake catalog is not well explained, only a paragraph is used, and resolved with "We made several runs to obtain coherence with the GR model and with the nominal slip rate." which seems like a trial and error until the results coincides with what we expected. This is not adequate for a serious research and the limitations and implications of the selected methodology to generate the synthetic catalog should be discussed.

The location of the events along the fault segments seems incoherent. As seen in Figure 5 there are areas of the fault with much higher accumulated slip, which means that there are sections with around 4 mm/yr while there are others with 2 – 3 mm/yr, which is tectonically inconsistent if the seismic coupling is 100%.

In general the method to generate the synthetic catalog is not well explained, neither discussed or justified. Being this part the most relevant of the work it is not acceptable.

The tsunami propagation code is not described, just cited as Miranda et al., 2014. In Miranda et al. 2014 the modeling does not include the Coriolis terms or bottom friction and no comparison with historical tsunami observations is done. I have doubts on the accuracy of the results as they are not compared to any observed tsunami in the area and the model is not conveniently described in the text.

Figures:

The figures in general needs to be improved.

Figure 1 I suggest the use of focal mechanisms representation for the three main events shown. This figure should be used also to present the tectonic setting with the main geological structures and the plate kinematics. The modeled structures (GF1, GF2 and GF3) could also be shown.

Figure 2 The synthetic catalog events cannot be distinguished. I suggest the use of a conditional symbol with the size of the circle proportional to the size of the modeled event. The modeled fault sections GF1, GF2 and GF3 should be shown.

Figure 3 The three events highlighted (stars) should be specified. Which one corresponds with the 1975 event? And with the 1941 event? The yellow color on a white background is not a wise decision.

Figure 4 The figure need to be reworked. The gray background and the extremely long titles of each subfigure should be changed. An a and b for each subfigure is recommended.

Figure 5 What is the vertical axis in the figures? Width? What are the parameters of the fault sketch? Are they used anywhere? If so they should be described in the figure caption, if not the sketch should be deleted.

Figure 6 This figure can be greatly improved, and/or simplified. On its present form is almost useless due to the accumulation of symbols and its similarity.

Tables:

In tables 2 and 3 the values of seismic moment should be adequately represented.

---

## Author Comment (AC1) · 30 Apr 2017

SEE FILE ZIP ATTACHED INCLUDING ANSWERS TO THE COMMENTS OF THE TWO REVIEWERS AND REVISED MANUSCRIPT

Please also note the supplement to this comment:
http://www.nat-hazards-earth-syst-sci-discuss.net/nhess-2017-57/nhess-2017-57-AC1-supplement.zip

---

## Author Comment (AC2) · 30 Apr 2017

SEE ZIP FILE ATTACHED INCLUDING ANSWERS TO REVIEWERS AND REVISED MANUSCRIPT WITH TRACK CHANGES

Please also note the supplement to this comment: http://www.nat-hazards-earth-syst-sci-discuss.net/nhess-2017-57/nhess-2017-57-AC2-supplement.zip